# Mechanical Resistance of New Apple Genotypes for Automated Harvesting

**DOI:** 10.3390/plants14223455

**Published:** 2025-11-12

**Authors:** Martin Císler, František Horejš, Jakub Lev, Petr Novák, Milan Kroulík, Lubor Zelený

**Affiliations:** 1Department of Agricultural Machinery, Faculty of Engineering, Czech University of Life Sciences Prague, Kamýcká 129, 165 00 Prague, Czech Republic; horejsf@tf.czu.cz (F.H.); kroulik@tf.czu.cz (M.K.); 2Department of Mathematics and Physics, Faculty of Engineering, Czech University of Life Sciences Prague, Kamýcká 129, 165 00 Prague, Czech Republic; jlev@tf.czu.cz; 3Research and Breeding Institute of Pomology Holovousy Ltd., Holovousy 1, 508 01 Hořice, Czech Republic; lubor.zeleny@vsuo.cz

**Keywords:** fruit evaluation, phenotype, fruit firmness, mechanical damage, initial flesh failure, breeding

## Abstract

Mechanical damage to apples that occurs without visible skin rupture represents a significant issue during handling and harvesting. The aim of this study was to evaluate the potential for detecting initial internal tissue failure using parameters derived from the deformation curve obtained during a quasi-static penetration test. Particular attention was given to the parameter Pa, defined as the pressure at the yield point, which indicates the onset of structural failure in the tissue. The occurrence of *Pa* was monitored across five apple genotypes, and the results demonstrated the parameter’s sensitivity to latent internal damage. The parameter *Pc*, characterising resistance in the later phase of penetration, also showed a strong correlation with fruit bulk density. Significant differences in all mechanical characteristics were observed between the genotypes. The results highlight the potential of these parameters for assessing varietal suitability for mechanised harvesting and sorting. The proposed methodology is readily applicable in the selection of new genotypes within breeding programmes as well as in post-harvest situations.

## 1. Introduction

Fruit represents an integral part of the human diet, and its consumption is important from both nutritional and health perspectives [1]. In the Czech Republic, apples are the most widely cultivated fruit, covering an area of 4832 hectares. In 2023, the apple production of professional orchards reached 100.60 thousand tonnes, with an average of roughly 110 thousand tonnes in recent years [2].

Only fruits that show no signs of internal or skin damage (e.g., bruising), have an intact stem, and meet defined quality criteria are classified as table apples of the so-called “Extra Class” [3]. Mechanical damage to fruit is a key factor affecting the quality of harvested produce and represents a major issue in post-harvest handling. Mechanical stress occurs at various stages of the process—from harvesting through sorting and packaging to transport and subsequent storage. The most common forms of such stress include impacts, vibrations, and compressive forces [4,5], as well as inappropriate storage conditions [6]. Losses due to mechanical damage are estimated to reach 40–50% of total production [7], with bruising identified as the primary cause [8]. The detection of bruises is particularly challenging, as they frequently occur beneath the fruit skin without external indicators [9]. Effective localisation of bruises is therefore complex, and its reliability depends on a wide range of variable factors, such as cultivar, stage of physiological maturity, and both pre-harvest and post-harvest handling conditions [10].

For the purpose of predicting the mechanical resistance of fruits, laboratory practice employs model simulations of mechanical loading, which allow for the quantification of cultivar-specific responses to defined stress stimuli [11]. A wide range of methods is used in experimental testing–among the most common are pendulum tests [12], free-fall methods [13,14], tensile tests [15], and compression tests [16,17].

Recent studies on the biomechanical properties of apples have mainly focused on quantifying bruise susceptibility, tissue firmness, and the energy thresholds associated with impact and compression loads [1,6]. Most of these works aimed to establish correlations between mechanical responses and fruit quality parameters, such as firmness, maturity, or internal structure. Advanced imaging and modelling techniques have also been employed to describe stress distribution within the fruit tissue and to simulate the onset of internal damage. Despite their contribution to the theoretical understanding of fruit mechanics, these studies provide limited insight into how mechanical resistance varies among genotypes and how such differences influence the suitability of apples for automated harvesting.

Identifying differences in mechanical resistance among individual genotypes requires sensitive and reproducible methods that enable the detection not only of damage occurrence but also of its characteristic progression [1]. Since bruises often develop without visible skin rupture, it is necessary to monitor the fruit’s response during loading with sufficient resolution to detect the onset of internal failure [18].

With the increasing effort to implement automated fruit harvesting and sorting systems, the issue of mechanical damage has once again come to the forefront. Although these technologies significantly improve harvesting efficiency, the interaction between robotic systems and fruit can lead to microtraumas without any visible external signs [9,19]. Such latent damage often becomes apparent only during storage and can critically affect the market quality of the fruit [1]. Paradoxically, new technologies thus highlight a problem that was previously addressed more intuitively during manual harvesting. The absence of natural sensory feedback (touch) in robotic systems necessitates the quantification of genotype-specific sensitivity.

This study presents a novel approach to assessing the mechanical resistance of apples, including parameters corresponding to the yield point, which can be used to predict fruit resistance and define safe stress limits for specific genotypes. The data obtained allow for the comparison of genotypes in terms of deformation, required force, and the nature of tissue damage, thereby paving the way for the targeted selection of genotypes suitable for modern technologies in automated harvesting and sorting.

## 2. Materials and Methods

The study focused on evaluating the mechanical properties of apple fruits. Testing was carried out on a total of five newly bred genotypes, with 9 to 24 fruits measured per genotype. The genotypes included B32 (18), HL648 (10 in the first and 24 in the second year), B217 (10), B231 (9), and B273 (9), with the number of tested fruits indicated in parentheses. The fruits were harvested manually at harvest maturity [20] from the orchard of the Research and Breeding Institute of Pomology Holovousy Ltd. Immediately after harvest, the fruits were placed into perforated plastic crates in a manner that prevented any mechanical damage caused by handling or transport to the workplace of the Czech University of Life Sciences in Prague. After harvesting, the fruits were stored for a maximum of five days in a cold chamber at 1 °C. This storage method was intended to stabilise the physiological state of the fruits [21].

For each tested fruit, basic morphological and physical characteristics were determined, including weight, dimensions, volume, and density. Fruit weight was measured using a Scaltec SBC 41 analytical balance (accuracy 0.01 g). Fruit height and two mutually perpendicular diameters in the equatorial region were recorded using a digital calliper with an accuracy of 0.1 mm. Fruit volume was determined using the liquid displacement method based on Archimedes’ principle [22]. Based on these measurements, fruit density was subsequently calculated as the ratio of its weight to its volume. The mean values and standard deviations of these measurements for all genotypes are summarized in Table 1.

The mechanical resistance of the fruits was evaluated using a compression test performed on the universal testing machine DEFORM 02 (PEMAR & Co. Ltd, Breclav, Czech Republic.). The device enables the measurement of force as a function of deformation up to 100 N and is designed for testing soft materials such as fruit.

Although the test setup provides accurate measurements of force and deformation, it should be noted that these parameters are empirical in nature, as the recorded values depend on the geometry of the tested sample.

Prior to measurement, the fruits were prepared by cutting each one longitudinally along the vertical axis. One of the halves was then placed flat-side down on the base of the testing device. Measurements were performed in the equatorial zone of the fruit, with the loading direction set perpendicular to the fruit surface. This region exhibits the most regular curvature and is frequently exposed to mechanical stress during handling and harvesting [6]. It was therefore selected for its geometrical stability and relevance to typical compression damage scenarios. The testing tool consisted of a brass cylindrical probe with a blunt end and a diameter of 8.1 mm, which was slowly pressed into the skin and subsequently into the flesh of the fruit during the test. The small probe diameter was chosen to minimise the effect of the gradual establishment of full contact between the probe surface and the fruit skin. Before the actual test began, the probe first approached the fruit surface at a higher speed, with a deceleration to the test speed occurring immediately after contact with the skin was established (at a force of 0.1 N). The loading itself proceeded at a constant speed of 0.05 mm s^−1^. This setting enabled detailed monitoring of the force–deformation relationship. The slow, controlled deformation process also allowed for the precise identification of key points on the resulting deformation curve. The test was terminated after reaching a penetration depth of 5 mm. A detailed view of the device and the tested fruit is shown in Figure 1.

The primary output of the compression test from the universal testing machine is a deformation curve, representing the relationship between deformation and compressive force. An example of a deformation curve is shown in Figure 2, illustrating the relationship for genotype B32. Before the onset of the linear region of the curve, a short non-linear segment appears, corresponding to the gradual establishment of full contact between the probe surface and the curved apple skin. The first part of the curve after full contact is approximately linear. In this region, no permanent damage to the fruit flesh occurs, and the deformation is elastic in nature. This section ends at the point *Fa*, beyond which structural failure occurs beneath the skin, resulting in permanent damage to the fruit (yield point). This region is fitted with a straight line, represented by a red dashed line in Figure 2. As deformation continues, the compressive force increases again until it reaches a maximum value marked as *Fb*. At this point, the skin is ruptured, and the testing probe begins to penetrate the fruit flesh. The compressive force then decreases and stabilises at an approximately constant or slightly declining value. The average value of this force is denoted as *Fc* in Figure 2 and is also indicated by a red dashed line. For the purposes of this study, the calculation of *Fc* was based on values recorded from 1 mm beyond the position of *Fb* until the end of the measurement.

For the purposes of further analysis, the forces *Fa*, *Fb*, and *Fc* were converted into pressure values, where pressure was defined as the ratio of force to the surface area of the testing probe. This yielded the parameters *Pa*, *Pb*, and *Pc*. In addition, the parameters *Ka* (the slope of the line fitted to the data from the beginning to point *Fa*) and *Kf*, representing the force ratio *Fa*/*Fb*, were determined. Fruit volume and density values were also included in the analysis.

For data processing and evaluation, the programming language Python (ver. 3.10, www.python.org, accessed on 16 July 2025) and the following supporting libraries were used: NumPy (ver. 2.1, https://numpy.org, accessed on 16 July 2025), Scikit-learn (ver. 1.6, https://scikit-learn.org, accessed on 16 July 2025), SciPy (ver. 1.14, https://scipy.org, accessed on 20 July 2025), and Matplotlib (ver. 3.9, https://matplotlib.org, accessed on 20 July 2025).

## 3. Results

The characteristic shape of the deformation curve is described in the previous chapter. Figure 3 compares all deformation curves obtained during the experiments. For each sample, compressive force values were recorded at 0.2 mm deformation intervals, and the mean force corresponding to each deformation point was plotted in the graph shown in Figure 3. The same graph also includes the standard deviation. The resulting curve represents a typical course of the deformation curve, with the exception of point *Fa*, which varies between individual samples and is therefore not clearly visible in the graph. Nevertheless, the graph clearly illustrates the approximate location of the maximum force value, the subsequent drop in compressive force, and the transition to a plateau or slow decline in force during penetration into the fruit flesh.

The course shown in Figure 2 and Figure 3 can also be divided into three distinct regions. The first is the region of elastic deformation, ending at point *Fa*, during which no permanent damage occurs. The second region begins at point *Fa* and ends at the breaking point marked in Figure 3. In this region, structural failure occurs first beneath the skin, followed by penetration through the skin. The final region represents the penetration of the probe through the fruit flesh.

Data on density, volume, and parameters extracted from the compressive force profiles were compared using box plots as illustrated in Figure 4, the Kruskal–Wallis test, and post hoc Dunn’s test. The Kruskal–Wallis test was chosen as some of the data did not meet the assumptions required for ANOVA (data normality). In all cases, significant differences between genotypes were observed. Subsequent Dunn’s tests revealed statistically significant differences (*p*-value < 0.05) in more than half of the cases. Of particular interest is genotype HL648, which was measured not only in 2023 but also in 2024 (HL648_24). For this genotype, no significant year-to-year differences were observed, except for the parameter *Ka*, where Dunn’s test resulted in a *p*-value = 0.0238. The highest values of bulk density were found in genotypes B231 and HL648. Similar results were observed for parameters *Pa* and *Pc*, suggesting a logical relationship between fruit flesh density and the compressive force required to damage it.

The relationships between the evaluated parameters required a more detailed investigation. Figure 5 presents a correlation matrix of all the assessed parameters. Histograms of individual parameters are shown along the diagonal, Pearson correlation coefficients are displayed above the diagonal, and graphical representations of the same values are shown below the diagonal. The strength of the correlation is indicated both by the size of the point and by the shade of its colour. Only statistically significant correlations (*p*-value < 0.05) are displayed. The data show that no significant correlation was found in only two cases. Very strong correlations were observed between *Pa* and *Kf*. However, this is due to the fact that *Kf* is defined as the ratio *Fa*/*Fb*, which already incorporates a dependency on *Fa*, and thus on the *Pa* parameter. A negative correlation between density and volume was observed, which likely reflects the influence of fruit size. Larger apples tend to contain a greater proportion of intercellular air spaces, resulting in lower flesh density [23]. Interesting, though not unexpected, is the strong relationship between both density and *Pc*, and volume and *Pc*. *Pc* represents the pressure exerted on the testing probe during penetration of the fruit flesh, indicating that fruit size and tissue density directly affect the resistance to penetration.

As the evaluated parameters were found to be strongly correlated, principal component analysis (PCA) was performed to further explore their interrelationships. Figure 6 shows the relationship between the two identified components, PC1 and PC2. Together, these two components explain 78.7% of the total variability (PC1—56.1%, PC2—22.3%). The individual genotypes are distinguished by colour, and clear differences between them can be observed. Interestingly, the differences are primarily driven by PC1, while PC2 appears to reflect variability within genotypes. Only in the case of B231 is a slightly lower PC2 value evident compared to the other genotypes.

Figure 6 also illustrates the influence of the evaluated parameters on components PC1 and PC2. This is represented by vectors whose components correspond to the loadings on PC1 and PC2. The results show that PC1 is primarily influenced by the parameters *Ka*, *Pc*, density, and volume, while PC2 is influenced by *Kf*, and to some extent by *Pa* and *Pb*, although these also contribute significantly to PC1. It appears that PC1 reflects properties related to tissue firmness and density, whereas PC2 reflects characteristics associated with the apple skin. This is because *Pa* corresponds to the pressure required to damage subcutaneous structures, *Pb* to the pressure needed to rupture the skin, and *Kf* to the ratio *Pa*/*Pb*.

## 4. Discussion

Most previous studies have focused primarily on the maximum force required to rupture the skin or the average penetration force, both of which reflect the later stages of fruit destruction. In contrast, the force or compressive stress corresponding to the yield point has so far been assessed only rarely. Yet, this aspect is crucial for understanding hidden damage, which significantly affects both the storability and sensory quality of fruit. The yield point marks the first disruption of subcutaneous cellular structures, devoid of any visible external signs. For example, Stropek and Gołacki [24] analysed the response of apple flesh to quasi-static and impact loading but concentrated mainly on the overall force required for fruit destruction. Similarly, Shirvani et al. [16] tested various elasticity models, but their attention was focused on global material behaviour rather than the point of initial damage.

The parameter *Pa*, detected at the yield point, provides a more accurate insight into the initial structural changes in the fruit flesh. This parameter demonstrates high application potential in the field of automated harvesting, where contact between the machine and the fruit often occurs without visible skin rupture. Davidson and Mo [19] highlighted that robotic handling can generate numerous microtraumas that are not detectable by standard inspection methods. The *Pa* parameter, therefore, enables a more precise definition of safe handling limits across different genotypes, which is essential for the design of harvesting systems and sorting technologies.

Among the findings of this study is the strong correlation between the bulk density of the fruit and its mechanical resistance to penetration, expressed as the pressure *Pc*, defined as the ratio of the mean resistance force to the surface area of the testing probe. This relationship indicates that the structure of the fruit flesh, and thus its density, plays a crucial role in the fruit’s ability to withstand mechanical stress. A similar relationship was described in the study by Chen et al. [25], where apple damage prediction was performed using a combination of experimental tests and FEM models. Winisdorffer et al. [26] further state that fruit damage resistance is largely determined by a combination of physical and chemical parameters, with higher density typically associated with a more stable cellular arrangement. Additionally, Cen et al. [27] demonstrated that increased porosity in fruits is often linked to lower mechanical resistance. The study by Lahaye et al. [28] also confirms that the water and pectin content in cell walls directly influence viscoelastic behaviour, which may explain the observed relationship between higher density and increased penetration resistance.

This study recorded significant differences among the tested apple genotypes in terms of their mechanical resistance and structural integrity. Stopa et al. [29] demonstrated variability in susceptibility to bruising among three commercial cultivars, with some showing damage even under relatively low loads, corresponding to the zone defined by the *Pa* parameter. Similarly, Stropek and Gołacki [24] found that these differences persist even after storage in controlled atmosphere conditions, suggesting the temporal stability of varietal characteristics.

This variability among genotypes also highlights the need for specific and sensitive methods for evaluating the mechanical properties of fruits. This relationship is further supported by the review by Li and Thomas [30], which points to the lack of a unified parameter for quantifying the onset of structural failure. Their conclusions emphasise the importance of separately assessing external and internal structures, such as the *Pa* parameter defined in this study, which specifically describes the properties of the skin and the underlying fruit tissue. Moreover, Ricci et al. [31] demonstrated that deformation profiles can also be used to predict sensory characteristics, further reinforcing the importance of evaluating specific genotypic traits during the initial stages of mechanical testing.

The main advantage of the proposed method lies in its simplicity and its ability to detect early, often latent damage to the fruit flesh. In contrast to complex computational approaches or visualisation technologies, such as FEM simulations or imaging techniques like MRI, the parameterisation of the deformation curve (particularly *Fa* and *Pa*) offers a tool that can be directly applied in breeding experiments, resistance testing of new genotypes, or in the development of harvesting and sorting technologies. In the future, this methodology could be extended using acoustic–mechanical sensors or correlations with chemical and optical characteristics of cellular structures, which could significantly enhance the predictive capacity for detecting hidden defects, before they become visually apparent.

Beyond its practical advantages, the method also provides valuable insight into the underlying mechanisms of tissue failure during localised loading. The observed mechanical response of apple tissue reflects the sequence of failure mechanisms occurring during localised loading. The yield point (*Pa*) indicates the onset of internal microcracking beneath the skin, whereas *Pb* corresponds to the rupture of epidermal and subepidermal layers. Beyond this point, the tissue gradually disintegrates during probe penetration (*Pc* phase), which is associated with the propagation of cracks and cell wall rupture in the parenchyma. The crack propagation mechanism may have a non-trivial influence on the force profile during penetration, particularly in relation to tissue texture and crispness. Although this phenomenon was not the main focus of the present study, it represents a promising direction for further research.

## 5. Conclusions

This article presents a test that enables the acquisition of comprehensive information on the mechanical resistance of apples within a single measurement. The derived parameter *Pa* represents the fruit’s resistance to bruising and thus provides information that may be crucial in the context of the rapidly developing field of automated harvesting, as it allows for the definition of safe load limits for harvesting tools.

The article evaluates new genotypes specifically bred with regard to their suitability for automated harvesting. In terms of resistance to bruising, genotype HL648 performed best, with a mean *Pa* value of 1.1 MPa. In contrast, the lowest value was observed for genotype B217 (*Pa* = 0.7 MPa).

The testing of new genotypes demonstrated the high sensitivity of the method to genotypic differences, confirming its relevance for the selection of new genotypes and the prediction of their resistance to damage. Due to its simplicity, the method represents a practical extension of standard fruit mechanical property tests, with the potential for integration into both breeding and post-harvest systems.

## Figures and Tables

**Figure 1 plants-14-03455-f001:**
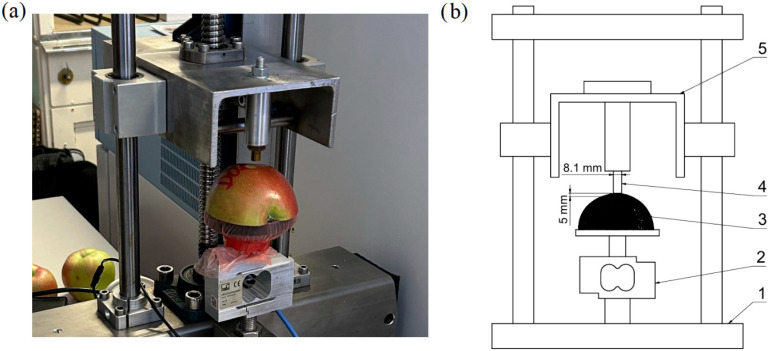
(**a**) Detail of the universal testing machine DEFORM 02 with the tested fruit. (**b**) Schematic diagram of the universal testing machine DEFORM 02. (1—main frame, 2—load cell, 3—specimen, 4—cylindrical probe, 5—press head).

**Figure 2 plants-14-03455-f002:**
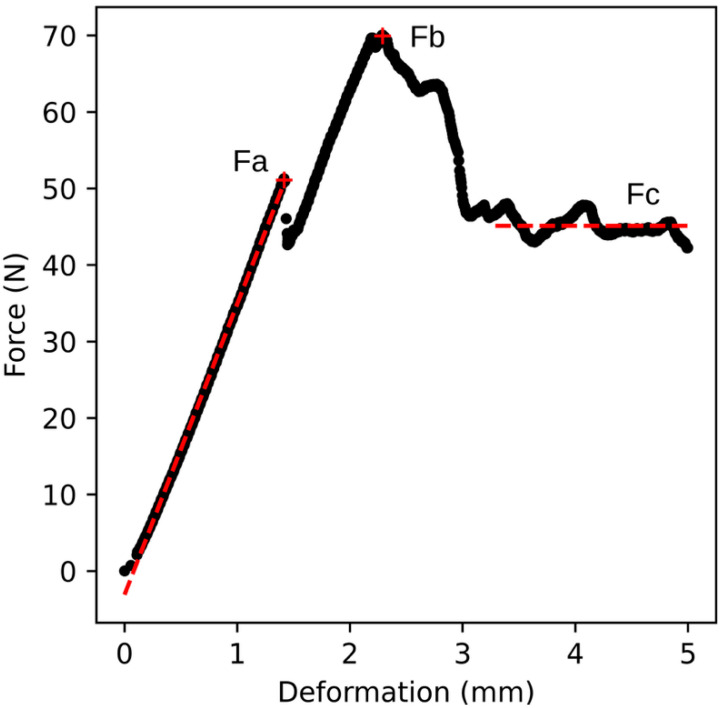
Relationship between compressive force and deformation for genotype B32 (sample no. 9). The figure shows the forces *Fa*, *Fb*, and *Fc*. The data from the beginning of the curve to point *Fa* are fitted with a straight line defined by the equation *y* = 38*x* − 3.1, *R*^2^ = 0.999 (red dashed line on the left). *Fc* represents the average force value in the final phase of deformation (red dashed line on the right).

**Figure 3 plants-14-03455-f003:**
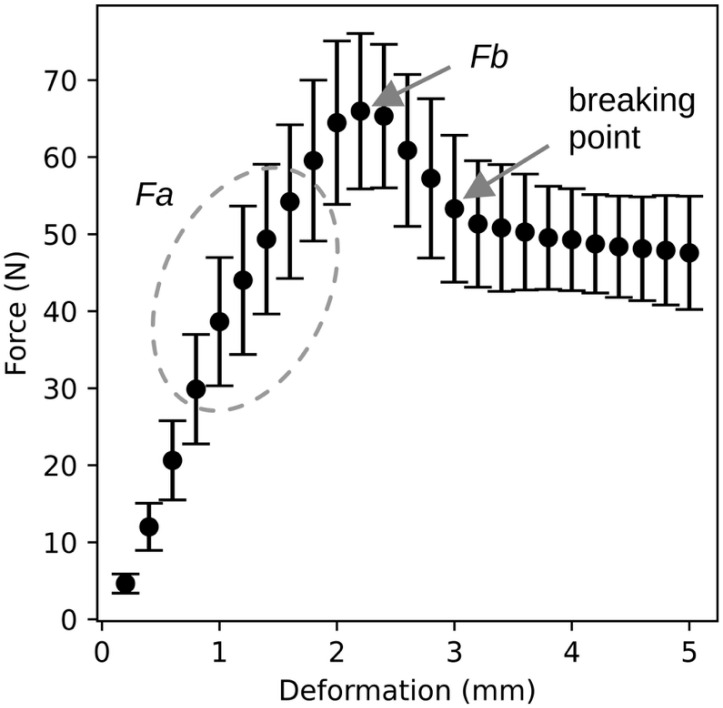
Comparison of the compressive force–deformation relationship for all measured samples. The graph shows the mean compressive force values for the corresponding deformation levels along with their standard deviations.

**Figure 4 plants-14-03455-f004:**
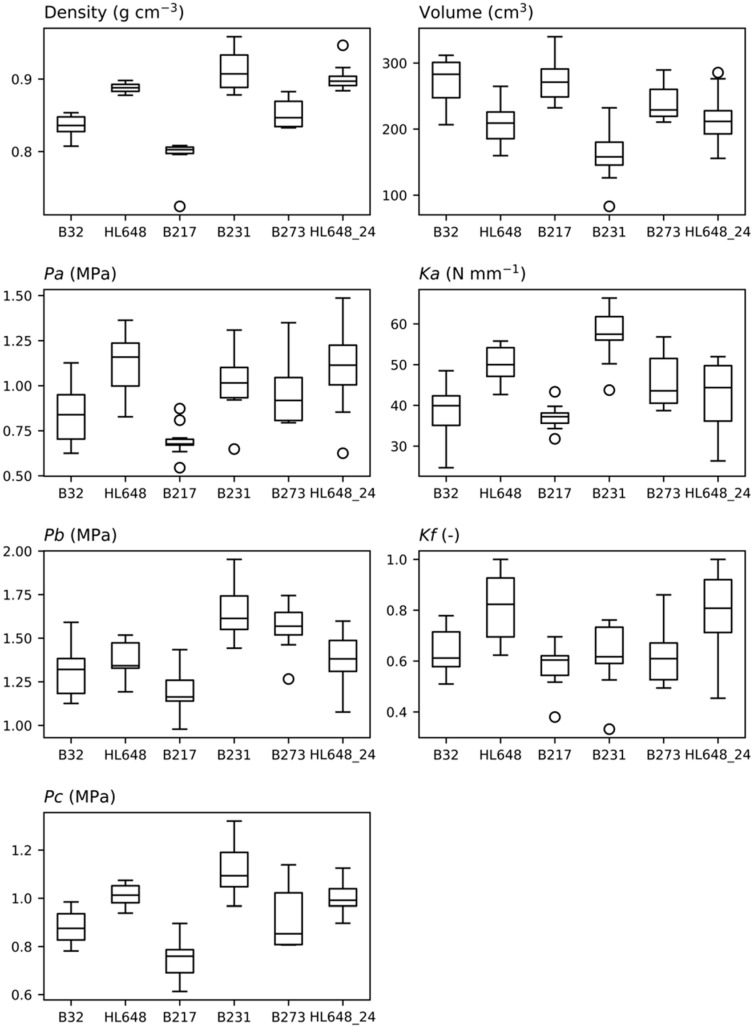
Comparison of the evaluated parameters between the tested genotypes using box plots. Median values, interquartile ranges, outside values, and outliers are depicted in the box plots.

**Figure 5 plants-14-03455-f005:**
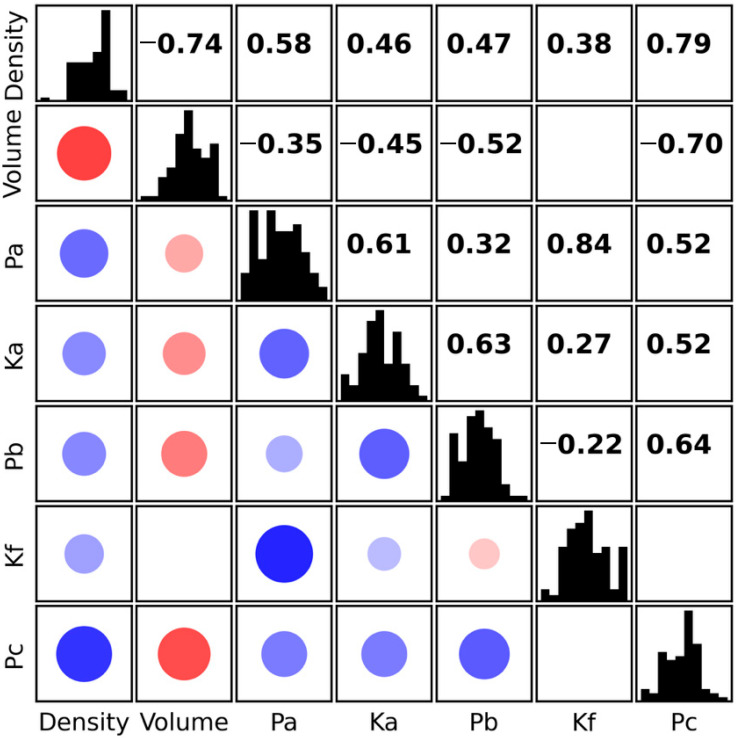
Correlation matrix of the evaluated parameters. Histograms of the parameters are shown along the diagonal; Pearson correlation coefficients are displayed above the diagonal, and graphical representations of the same values are shown below the diagonal. The strength of the correlation is indicated by both the size of the point and the shade of its colour. Positive correlations are shown in blue, negative in red. Only statistically significant correlations (*p*-value < 0.05) are displayed.

**Figure 6 plants-14-03455-f006:**
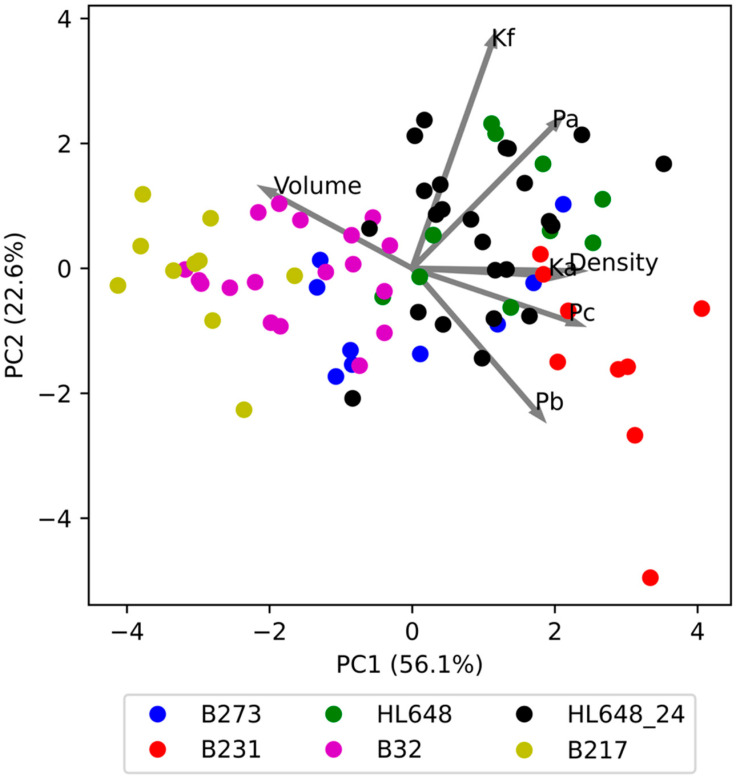
Graphical representation of principal component analysis for the evaluated parameters, including loadings for PC1 and PC2 (loadings values multiplied by five for improved readability).

**Table 1 plants-14-03455-t001:** Summary of the measured basic morphological and physical characteristics of all apple genotypes examined. Average values and standard deviations (SD) are presented for fruit mass (*m*), height (*h*), two equatorial diameters (*d*_1_, *d*_2_), volume (*V*), and density (*ρ*).

Genotype	*m* (g)	*h* (mm)	*d*_1_ (mm)	*d*_2_ (mm)	*V* (cm^3^)	*ρ* (g/cm^3^)
	SD *m*	SD *h*	SD *d_1_*	SD *d_2_*	SD *V*	SD *ρ*
B32	227.4	71.6	79.5	81.5	268.0	0.85
	28.6	4.9	3.9	4.3	35.7	0.08
HL648	188.7	70.7	71.3	71.7	213.3	0.89
	22.2	7.0	2.7	3.1	26.1	0.01
B217	218.6	66.9	83.5	82.3	270.5	0.80
	25.7	3.9	3.8	3.1	34.7	0.02
B231	148.7	56.2	69.3	69.4	159.9	0.93
	34.5	5.7	6.2	5.8	34.9	0.13
B273	208.5	68.1	78.2	79.1	239.7	0.88
	25.2	6.5	4.8	3.4	38.4	0.12
HL648_24	198.7	68.4	73.6	74.9	220.1	0.90
	27.6	4.4	5.2	3.6	30.4	0.02

## Data Availability

The data that support the findings of this study are available from the corresponding author upon reasonable request.

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
