# Peer review of "Mechanical Resistance of New Apple Genotypes for Automated Harvesting"

_plants, 2025, doi:10.3390/plants14223455_

Round 1
Reviewer 1 Report
Comments and Suggestions for Authors
On the first page. Delete the Institution marked as 3. None of the authors belong to that institution
In section Methods consider that mechanical tests using an equipment as the one described, provide accurate measurements of force and deformation, but they are empirical because the recorded values depend on the sample geometry.
For that, in the methods section (line 91), provide a table of the mass, diameter, volume, and density values with their standard deviation for all genotypes studied.
Their Pa results are expressed as pressure (Force/area); however, the contact area with the probe, changes throughout the test (at least until the entire probe surface makes full contact with the skin of the sample). This will be critical if the sample diameters were different.
It is important that the authors clearly show how Pa was not significantly influenced by sample size and if this is the case, provide a further explanation to support their conclusions.
I agree that fruit flesh density is an important factor in the mechanical properties of apples, but this property changes with fruit size. Because of the gases ocluided into apple tissue. The authors should explain their results further in the context of sample size and diameter
Author Response
Viz pÅ™íloha

Reviewer 2 Report
Comments and Suggestions for Authors
The manuscript has conducted puncture mechanical property tests on different types of apples and carried out relevant analyses. The research conclusions can provide data support for the mechanical design of apple mechanical harvesting and other applications. It is suggested to supplement or revise the following issues:
1.In the Introduction section, the authors are requested to supplement a detailed discussion on the methods and results of apple biomechanical tests at the current stage, and must highlight the targeted nature and innovation of this study.
2.In the Materials and Methods section:
(1) The manuscript mentions cutting apples for testing. However, as an anisotropic biological material, apples have significant differences in mechanical properties across different parts of the fruit. What are the specific rules for selecting test points on half of the apple? And can these test points represent the mechanically damaged areas?
(2) As a biological material, apples undergo mechanical property tests during 5 days of storage. During this period, the physiological state of apples (e.g., moisture content) has a great impact on their mechanical properties. Is there any potential impact of this factor? Please consider this issue.
(3) Generally, the mechanical harvesting process of apples may involve damages such as collision and extrusion. Why is only the puncture test selected to measure the relevant mechanical property indicators? Moreover, there is a large difference between quasi-static tests and actual harvesting conditions. Please elaborate on the necessity of selecting the puncture test in detail.
3.In the Results and Discussion section: What are the damage mechanisms of apple peel and pulp reflected by the damage mechanical properties of apples? Is it considered to further elaborate using the crack propagation mechanism and microscopic images?
Round 2
Reviewer 1 Report
Comments and Suggestions for Authors
The observations were adequately addressed and the new version of the manuscript is technically better
Author Response
Comments 1: The observations were adequately addressed and the new version of the manuscript is technically better.
Response 1: We would like to thank you sincerely for your careful review and constructive feedback.
We truly appreciate your positive assessment that the revised version of the manuscript is technically improved and that the previous comments were adequately addressed.
Thank you for taking the time to evaluate our work and for your helpful input.
Reviewer 2 Report
Comments and Suggestions for Authors
The author has made revisions to the original manuscript regarding this issue and suggests that corresponding references be added to lines 115-119 to support this inference.
Author Response
Comments 1: The author has made revisions to the original manuscript regarding this issue and suggests that corresponding references be added to lines 115-119 to support this inference.
Response 1:
Dear reviewer,Thank you for your comment. We have, of course, added the missing citation. After consulting with our colleagues, we have decided to slightly modify our statement (lines 115-119) to better reflect our motivation for selecting the area for measurement. "it was therefore selected to minimise the influence of anatomical anisotropy on the measured mechanical response." is modified to: "It was therefore selected for its geometrical stability and relevance to typical compression damage scenarios."